# Water Disconnection and Vital Flow Policies: International Practices in Medium- and High-Income Countries

Rui Cunha Marques [1] , Pedro Simões [1], Inês Machete [1] and Thalita Fagundes [1,2,]*

1 Civil Engineering Research and Innovation for Sustainability (CERIS), Instituto Superior Técnico, Universidade de Lisboa, Av. Rovisco Pais, 1049-001 Lisbon, Portugal
2 Department of Hydraulic and Sanitation, University of São Paulo, Avenida Trabalhador São-Carlense 400, São Carlos 13560-970, SP, Brazil
* Correspondence: thalita.fagundes@tecnico.ulisboa.pt

**Abstract:** Water access is recognized as a human right by the United Nations since 2010. However, even when piped water is available, the economic crisis has limited poorer households to afford those services on a regular basis. Users become debtors as utilities face growing service costs and shrinking national public funds, pushing tariffs as the main source of revenue for cost recovery. The COVID-19 pandemic lit up affordability issues and health consequences of disconnection policies. Countries worldwide banned water shutoffs during the last year, ensuring water and wastewater service access for all citizens. Although disconnection is a way to reduce the number of debtors, it is, at the same time, considered a threat to human rights statements. This study gathered information on water subsidies, disconnection and vital flow policies applied among several medium- and high-income countries, in order to analyze how utilities have dealt with default rates and if there is any difference between the approaches between medium- and high-income countries. Through case studies, this paper also aims to inspire other practitioners facing the same issues. Based on the eleven case studies presented here, we concluded that high-income countries use assistance programs over water disconnection policies when compared to medium-income ones. Water shutoffs are explicitly forbidden in the United Kingdom, Australia, and France. Although a humane alternative, water flow restrictors have limited application, especially considering the technical issues involved.

**Keywords:** water access; water disconnection; default; vital water flow

## 1. Introduction

The costs of water supply and wastewater services (WWSs) have risen faster than the average income to overcome aging infrastructure in high-income countries and the lack of coverage in low- and medium-income ones. Affordability has, then, become a burgeoning concern for governments and academics, since vulnerable families have struggled to keep payments on time [1,2]. Simultaneously, the decreasing availability of the public budget is converting tariffs into a key instrument for maintaining the quality of service.

Poorest households are even more prone to face difficulties in basic service access. A study with 35 European countries identified that in 2018, people at risk of poverty experienced a higher level of arrears on utility bills than the overall population: 16.3% compared to 6.8% [3]. WWS providers have dealt with affordability issues and default rates, with various approaches according to the type of service, customer socioeconomic aspects, and national or local regulation and public policies.

Subsidies for utility services are a widespread attempt to shrink the access gap in the water supply, wastewater, electricity, and gas sectors. The improvement of the social welfare of the poor by facilitating their access to such services is one of the main motivations to apply subsidies. There is a vast variety in the forms that such subsidies can take, and the details of their design determine the targeting performance [4–6]. Water utility subsidies

can be designed as consumption or connection ones, and also as targeted or untargeted [7]. Consumption subsidies usually operate through the tariff structure, by reducing the real cost of the service for all or for a group, but also through cash transfer to reimburse households' utility expenditures. Connection subsidies, on the other hand, are focused on non-connected families and aim to raise coverage rates, reduce connection fees, or help housing improvements. Both can be targeted or untargeted. If the subsidies are designed to reach all customer base, underpricing the service for every user, then they are considered untargeted. In practice, targeted and untargeted subsidies are often combined, since there may be a subsidy for capital expenses for example, but also a greater discount applied for specific users [7]. Subsidy policies vary among countries and localities. In two-thirds of the 35 countries covered by a recent study in Europe [3], at least one type of measure was identified: reduced tariffs, cash benefits, in-kind benefits, or basic/uninterrupted supply of water, aimed at helping low-income households to access WWSs. Reduced tariffs and cash benefits are by far the most common types of support measures observed in Europe [3] and in South America [4]. The authors also observed that most European countries use regional- and/or local-level support measures to facilitate WWS access, since such services are mostly organized at the subnational level.

Some researchers have warned about the performance of water and wastewater subsidies [7–9]. If not well targeted, they may not reach the poorest families, and consequently their main goal; at the same time, it can undermine the utility financial situation, provoking declining service quality. Research about subsidy performance concluded that the most common subsidy instruments, such as those delivered through increasing block tariffs (IBT), perform poorly in comparison to other transfer mechanisms [7]. Alternative consumption and connection subsidy mechanisms show more promise. Another study conducted in Cape Verde, Rwanda, and São Tome and Principe, also concluded that subsidies for electricity through IBT tariff structures were regressive due to a lack of access that prevents the poor from getting the services, and that connection subsidies would be more efficient [10].

As mentioned, poorly targeted subsidies do not reach—or reach at a very limited extension—vulnerable families [5–7]. Default rates then become an important issue for utilities, impacting their financial situation and slowing down expansion and maintenance planning. Some localities are not able to decrease default rates even with subsidy policies due to users' complete inability to pay for essential services or an unwillingness to pay for WWSs [11,12]. Water providers worldwide have then implemented disconnection policies towards shrinking users' debt, forcing delinquent customers to pay their outstanding bills.

Although widely spread, disconnection policies have been questioned by practitioners, since WWS access is recognized as a human right by the United Nations. Evidence from Europe, for example, reveals that the human right to water is only fully protected in 11 out of the 27 EU Member States and in the UK, where there is a clear rule for an uninterrupted basic supply of water [3]. A study of American water utilities suggested fifteen million Americans experienced water shutoffs in 2016, since most states lack statutory protections against them [13]. The authors observed, though, that shutoff provisions are not enforced evenhandedly, since the number of households eligible for disconnection varies from those who experienced a shutoff. An analysis of a 2015 national survey of 1897 American municipalities found a positive correlation between publicly owned water utilities and policies to protect low-income residents from disconnection [14]. On the contrary, no major differences between public and private providers in terms of institutionalized solutions for affordability, such as social and large family tariffs, and interruption proceedings regulation were observed in Portugal [15].

Utilities spend money on unsuccessful efforts to collect missing revenue, including the administration and enforcement of shutoff provisions, increasing operating costs. The implementation costs incurred by the system and the household, such as reconnection, along with the cost to public health and public trust, may be greater than the revenue recovered by water shutoffs [16]. Yet, the default rates remain an issue for water providers. The author mentions that utilities may be able to address affordability through programs

and prices if they overcome political, legal, and financial barriers. Although assistance programs can be complicated and costly to administer on a large scale, adjusting at a household level, it has been used across the globe to surpass users' inability to pay. Income-based water affordability programs have the potential to protect customers from disconnection while improving revenue stability, as observed in Philadelphia, U.S.A. through the Tiered Assistance Program, and in Baltimore, U.S.A. through the Water Accountability and Equity Act, where they cap water rates at a specific percent of household income for poor customers [13].

A study conducted by the World Bank in Uganda analyzed how some of the called pro-poor policies increased water coverage in Kampala's informal settlements, since the majority of poor people resided in those areas [17]. The practices applied included (i) an affordable connection fee for any user living within 50 m of water mains, (ii) a different tariff for different customers, (iii) specific subsidy for connection in poor settlements, including shared yard taps and prepaid Public Water Points (PWPs). The authors concluded that in general, the pro-poor program contributed to the expansion of services to poor households, but that disconnections were an issue for 21% of yard taps and 53% of PWPs.

The socioeconomic crisis, enhanced by COVID-19, has affected the poorest households worldwide even more, raising utility default rates. Since WWSs are one of the most essential services to avoid COVID-19 spread, many governments banned water disconnection during pandemics to ensure access to basic hygiene for every citizen during that difficult time [13,14].

Even though the prohibition of service disconnection affects utility financial health, and may not contribute to water preservation, WWS access is recognized as a human right. The American Water Works Association recommended water shutoff protections during the pandemic [18]. A study investigated water disconnection moratoriums during the COVID-19 pandemic in the U.S.A and observed that states which had economic regulation of private water utilities were more likely to impose moratoriums, and those with higher COVID-19 case rates imposed moratoriums earlier. Localities without statewide moratoriums were more likely to impose moratoriums if they had higher income, more minority residents, and more income inequality, since they had both the need and capacity to act [18].

Research examined the link between water access, COVID-19 daily infection and death growth rates in the U.S.A. [19]. The authors concluded that forbidding water disconnection significantly lowered the COVID-19 infection daily growth rate by 0.235%, and significantly lowered the death growth rate by 0.135%. Since the pandemic has worsened income loss, and the study showed that states with a higher percentage of minorities and essential workers had higher COVID-19 daily infection and death growth rates, the link between water affordability and the impact of COVID-19 on minorities and poorer families led the researchers to recommend protection from water shutoffs across states, especially for vulnerable households [19].

Therefore, water providers currently face the challenge of decreasing users' debt, without jeopardizing their human right to basic services. This study aimed to identify and discuss the strategies adopted by countries with different socioeconomic contexts to preserve water access for everyone, while maintaining the financial and economic sustainability of utilities. The research gathered the main international practices, including subsidy types, utilities commercial rules, and default policies applied, especially disconnection and minimum flow approaches. The present paper contributes to the literature by bringing and discussing WWS initiatives to overcome the nonpayment issue, ensuring water access and companies' financial health. It may help decision-makers and be an inspiration for practitioners to adapt one or some of the approaches mentioned here to local realities. The study found that most of the countries allow water disconnection, with a previous warning to the user. Subsidies are present in every country studied with different forms, considering the free monthly minimum quantity of water for general aid through broader social

funds. The research also perceived that the use of water flow restrictors as an alternative to disconnection is very limited and faces technical challenges.

This paper is organized as follows: after this brief introduction, Section 2 describes the research methodology applied. Section 3 presents and discusses the main practices addressed by the case studies. Section 4 provides the concluding remarks and suggestions for further research.

## 2. Materials and Methods

The present study compared international practices among countries worldwide, regarding the social policies to ensure the human right to WWS access, the approaches to deal with debtors, including disconnection, and the vital water flow policies adopted. The aspects analyzed include the topics: WWS coverage, average services tariff, default indicator, WWS access public policies, social aid to vulnerable households, disconnection practices with details, and restrictions of water flow as an alternative.

The case studies represent different socioeconomic contexts, including different continents, medium- and high-income countries: Australia, Brazil, Colombia, France, Germany, Italy, Mexico, Portugal, Spain, Singapore, and the United Kingdom. The study was based on official documents and interviews with the main stakeholders, such as utilities, water associations, and regulatory agencies. The purpose of the interviews was to confirm the data found online and gather complementary information, used in the following discussion.

## 3. Results and Discussion

The case studies are displayed in Figure 1. As mentioned in the previous section, we based this discussion on eleven case studies, taking into account different socioeconomic contexts, including WWS coverage rates.

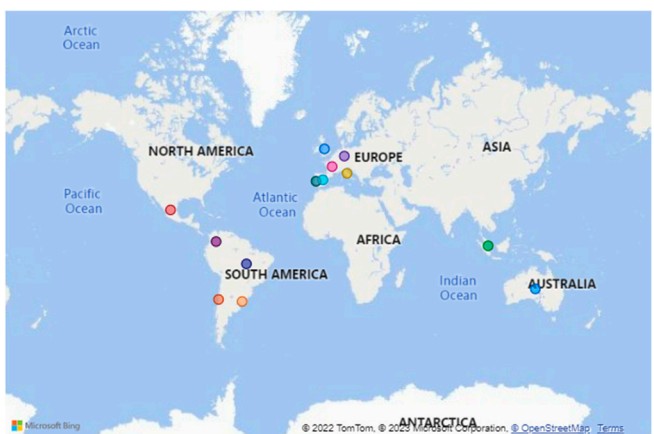

**Figure 1.** Case studies.

### 3.1. Default Rates

Table 1 summarizes the main characteristics of the countries related to WWSs, including nonpayment. As can be seen, three of the countries are classified as medium-income ones and the infrastructure services are not universalized yet. Water supply and wastewater follow the same path, where wastewater service coverage barely reaches 60% in Brazil, leaving around 95 million habitants without treated wastewater. Alongside that, water supply is not reaching all population in those Latin-American countries, yet. On the other hand, the eight countries considered high-income nations, have coverage rates of around 100% for water supply and 90% for wastewater.

**Table 1.** Case study main characteristics.

| Case Study | GDP Per Capita (USD, 2022) [2] | Potable Water Coverage | Wastewater Coverage | Average Consumption (L/capita·day) | Average Water and Wastewater Tariff (USD/m$^3$) | Non-Revenue Water | Default Rate |
|---|---|---|---|---|---|---|---|
| Australia | 60,443 | 98% | 90% | 162–373 | 2.53 | 10% | <1% |
| Brazil | 7507 | 84% | 54% | 154 | 1.92 | 39% | 2.50% |
| Chile | 16,265 | 100% | 97% | 161 | 2.12 | 33.4% | 6.5% |
| Colombia | 6104 | 86% | 75% | 145 | 1.58 | 40% | 5% [1] |
| France | 43,659 | 100% [1] | 82% | 170 | 3.35 | 20% | 1.90% |
| Germany | 51,204 | 100% [1] | 97% | 127 | 6.65 | 6% | <1% |
| Italy | 35,657 | 100% [1] | 85% | 132 | 2.51 | 43% | 1.6%—North 6.0%—South |
| Mexico | 10,046 | 94% | 77% | 202 | N/D | 51% | 30% of the users do not pay on time |
| Portugal | 24,567 | 99% | 85% | 189 | 2.42 | 29% | 0.7–1.4% |
| Singapore | 72,794 | 100% [1] | 100% [1] | 141 | 2.06 | 5% | 0.50% |
| Spain | 30,103 | 100% [1] | 89% | 128 | 2.29 | 23% | 1–2% |
| UK | 46,510 | 100% [1] | 100% [1] | 142 | 4.31 | 19% | 2% |
| Uruguay | 17,313 | 99.5% | 99% [3] | 134 | 2.55 | 52% | <5% |

[1] Approximately. [2] Source: Banco Mundial, 2021: https://data.worldbank.org/indicator/NY.GDP.PCAP.CD?name_desc=false (accessed on 10 January 2023). [3] Septic system and sewage.

For this research, coverage rates are not important just to visualize the local WWS access. Within the funding policies context, it is crucial to understand how far the utilities' role could go through subsidies, and who will most benefit from these practices. As mentioned before, water providers can be responsible for some types of subsidies, including the ones from tariff structure and connection fees [4,5,7]. Cross-subsidization is one common practice among WWSs. The most common way is charging wealthier groups more than the actual cost in order to reduce the price for the vulnerable population, through social tariff and IBT structures. IBT divides consumption levels into blocks, with increasing unit prices applied per each block. The unit price is lower on the first blocks, in an attempt to provide the minimum volume of water at an affordable price. The underpriced blocks are often subsidized by the higher consumption ones, and the assumption here is that richer households consume higher volumes, which can be false in many cases, such as larger families [5,6,10,20,21]. In 2018, half of the water utilities covered by the IBNET database and survey conducted by Global Water Intelligence used IBT schemes, especially in Latin America (70%), Middle East and North Africa (74%), and East Asia and the Pacific (78%) [6]. Along with local socioeconomic factors, a high coverage rate may provide a more propitious environment for utilities implementing subsidy policies through tariffs.

High-income obviously has the opposite influence on default rates since the population can afford essential bills. This is reflected in the water sector as can be seen in Table 1, where countries with a higher gross domestic product (GDP) have lower default rates, even with higher water and wastewater tariffs. Except for the south Italy, the default rates do not surpass 2% in high-income countries. The lower default rates and high GDP also contribute to a propitious environment for subsidy policies for those in need, since the utility has revenue from the tariff for the service itself and for social programs. Data for Mexico shows that only 70% of customers pay their bills on time. Although clearly important, GDP is not the only explanation for low default rates, as it can be observed in France and the United Kingdom, for example [22]. Those rates may also be related to the socio-cultural behaviors and the relationship the local population has with water resources. It is also a reflection of water conservation policies applied, as we discuss further.

Utilities can improve water access by poor households through network expansion and affordable service. The affordable price depends on various factors, including socioeconomic local reality, but efficient providers manage to lower operational and maintenance costs, reflecting positively on users' budgets. The best technical approaches to improve utility efficiency is not the subject of this paper, but it includes a decrease in non-revenue water indicators, which in turn, have default rates as one of the components. During the

COVID-19 pandemic, the essential role WWSs have on families' health and that depriving the poor of these services is extremely unwise became clearer.

### 3.2. Assistance Programs

Subsidy policies in the water sector embrace a huge variety of forms, from tariff structure to capital funding or general cash transfer. One of the most common is infrastructure construction with tax transfers, usually from the national budget. The other prevalent way is cross-subsidization, implicitly present in the IBT structure and very common in Latin America, or discounts for vulnerable households, such as social tariffs, punctual support for unpaid bills due to temporary unemployment, and connection fee alleviation [4–6,20]. The fact that in the IBT structure the subsidies reach every user who consumes within the first block, and do not consider (in the majority of the cases) larger families, makes this type of tariff regressive [6,7]. In many countries, fiscal incentives for the sector, such as reduced taxes, are in place to reduce operational costs, and consequently users' tariffs. Additionally, international development banks have applied lower interest rates to infrastructure projects as a way to subsidize and propel the sector.

One of the most common practices worldwide to improve families' affordability—for those with physical access to infrastructure networks—is the application of social tariffs, a subsidized tariff applied for vulnerable households [23,24]. Table 2 shows the prevalence of social tariffs among the case studies, together with other social programs applied.

**Table 2.** Case study assistance programs.

| Case Studies | Existence of Social Tariffs | Existence of Other Social Programs |
| --- | --- | --- |
| Australia | No | Yes. However, not specific to WWS |
| Brazil | Yes | No |
| Chile | Yes | Yes |
| Colombia | Yes | Yes |
| France | Yes | Yes |
| Germany | No | Yes. However, not specific to WWS |
| Italy | Yes | No |
| Mexico | Yes | No |
| Portugal | Yes | No |
| Singapore | No | Yes. It applies to specific poor areas. |
| Spain | Yes | Yes |
| UK | Yes | Yes |
| Uruguay | Yes | Yes |

As can be seen among the case studies, only Germany, Australia, and Singapore do not have social tariff schemes [25–27]. According to Table 1, those countries have default rates lower than 1%. Even though these three countries have a high GDP, they also use social assistance not as a specific water and wastewater program, but as a broader program, of which the budget comes from general tax.

Germany is one example of heavy government participation during the universalization process. Although German water utilities are self-sufficient now, the government provided heavy support in the past for wastewater coverage and still makes use of general social funds to support vulnerable families. In some cases, local welfare authorities even pay the poorest families' bills straight to utilities [25].

Brazil applies different approaches to the water sector. Through the Ministry of Health, Ministry of Environment, and Ministry of Regional Development, the country has implemented big national projects to boost water and wastewater infrastructure, funding network expansion with general taxes. The country also applies social tariffs in every state, although the effectiveness of the program is not considered adequate enough yet, and most users must claim the benefit [28]. The Brazilian Development Bank (BNDES) has also been implementing lower interest rates for water providers and other utilities to promote the infrastructure sector.

Chile has national policies assuring access to a minimum level of services of WSS for vulnerable families. The Law no. 18.778/89 states a direct subsidy for WSS tariffs up to 20 m$^3$ consumed for families unable to afford their bills [29]. The reduction varies between 25% and 85% of fixed and variable tariff components. The Chilean government also runs other general support programs, such as Chile Solidario, where the water bill full payment is included in the benefits for extremely poor families [30].

Uruguay also has more than one approach to guarantee water access to the poor. The national provider applies a fixed price of up to 15 m$^3$ consumed as a social tariff. The country also offers 100% WSS bill payment for poor, retired people, up to 10 m$^3$ consumed, and a different and fixed charge for small rural communities [30].

The Spanish state also covers a great part of the investments in the water and wastewater sector. For 93% of Spain's territory, there is at least one social mechanism available to help vulnerable households, either social tariffs or social funds [31]. The criteria to access those are established locally, and the budget comes from general tax (social funds) and tariffs from other users (cross-subsidization). Regarding France, the Housing Solidarity Funds, Fonds de Solidarité Logement (FSL), financially assists people who struggle to pay any housing-related bills on time. Moreover, the association of the water sector (Fedération Professionnelle des Entreprises de l'Eau, FP2E) has been strengthening social policies, such as increasing contribution to FSL, promoting social tariffs among utilities, and paying water bills for vulnerable families [22]. One of the goals of FP2E is to keep the burden from WWS at a maximum of 3% of families' income.

Italy does not face a coverage problem, as noticed in Table 1, but to avoid affordability issues, a specific benefit, called a "water bonus", was introduced in 2018. Italian utilities must offer 50 L per capita per day (lpcd) at no cost to the poorest families with an income less than USD 10,000 a year. This amount is considered basic for personal needs, and it is deducted from water consumption in the water bill [32]. The benefit is automatic, which means the families do not need to sign up for it. The subsidy is paid through cross-subsidization [3].

The Portuguese water regulator (ERSAR) recommends the implementation of social tariffs, eliminating the fixed component of the water tariff structure, and also raising the first water block (IBT) from 5 to 15 m$^3$ [24]. The budget for implementing the initiative comes from municipalities and, despite it is not being mandatory, 85% of Portuguese cities have implemented it. ERSAR also recommends specific pricing for larger families (household size larger than four), which includes adjustment of IBT consumption limits depending on the household size [33,34]. In addition, the Portuguese government finances a great part of investments in the water and wastewater sector, relieving the ending price.

Social tariffs in England and Wales vary among utilities and have been implemented since 2013. In general, the budget comes from cross-subsidization. Yorkshire Water is an example of a provider offering a variety of social programs to better address the specific problems vulnerable households may face. The programs go from regular support, such as WaterSure, WaterSupport, and Community Trust, to one-time assistance, such as Payment Holiday. WaterSure reduces poorer households' bills, if metered, and if the family has at least three children or relatives with a medical condition that requires a lot of water. WaterSupport offers vulnerable families' bill payment when it surpasses a certain amount. The Community Trust program pays delayed bills if the family has another priority debt at the same time, such as renting or local taxes [35–38]. The Payment Holiday, for instance, is made for users whose financial situation has dramatically changed (an unemployment situation, for example), and the program applies tariff discounts or even suspension of charges. In Scotland, vulnerable families can get a discount on water bills up to 35%, financed by municipalities. In the case of North Ireland, residential users do not pay for WWSs, as the budget for those services comes from general taxes.

Australia, for instance, does not apply social tariffs. Poorer families get support from general social funds. The provider Sydney Water, for example, helps vulnerable clients to find out the social assistance they have the right to, and provides more suitable payment

plans [26]. Colombia divides the population into six socioeconomic strata according to their housing conditions. The discounts are defined by local councils, but they can go up to 70% for the poorest group, which is stratus 1. The budget for this subsidy comes from the municipal general budget and cross-subsidization, where stratus 5 and 6, commercial and industrial clients pay more [30]. Mexico also uses tariff subsidies as an economic instrument to address default rates and vulnerable families' right to WWS access. Many Mexican cities apply social tariffs, and the methodology and discounts are set by each municipality [39]. Mexico City, for example, applies different tariffs, according to users' classification, which in turn considers housing typology [40].

As observed by the examples, social programs and policies vary greatly between localities. Some countries face water affordability within the broader poverty issue and fight it as an income problem, as seen in Germany, Australia, and Singapore. On the other hand, in some countries, water and wastewater utilities have a bigger role in assuring WSS access to poorer families, as seen in the range of social programs offered by them in the case studies. Pragmatically, utilities are service providers, and they should have a limited role regarding WSS affordability, such as working as efficiently as possible, which will be translated into lower tariffs and expanded WSS coverage rates and service quality to all users, despite their economic status. Sector-regulators play an additional important role, ensuring that providers are performing with high quality for every single person. Governments should be the main responsible establishment for fighting poverty, and this includes WSS access and affordability. Social programs towards income increase should be designed for providing a normal living condition to poorer families, including the ability to pay for WSS. All those entities, respecting the local singularities, could work together to optimize the public budget and sector subsidies. While this integrated and broader-welfare assistance approach is not implemented, utilities still have a fundamental role in guaranteeing WSS access to vulnerable families through social programs, such as the ones mentioned in this section.

### 3.3. Disconnection and Vital Water Flow Policies

The variety of subsidy policies noticed during the research reflects the influence socioeconomic and cultural factors have on water public policies. Although a great range of assistance programs was observed in all countries studied in this paper, default rates may persist, bringing in financial issues to utilities. Providers have implemented more drastic alternatives, such as service disconnection. With exception of three, all case studies allow water shutoffs as a way to recover users' debts (Table 3). With moratoriums on water shutoffs placed by many cities during the COVID-19 pandemic, the search for alternatives less severe methods had increased. The flow restrictor is an example of a humane alternative to absolute disconnection, although it requires investment in technology and labor, as well as engagement with engineering professionals [16]. The equipment has been applied in a few places and has been tested in others to provide at least a vital water flow for debtors. Table 3 shows how the case studies have been working with disconnection practices and flow restrictions in an attempt to decrease default rates.

During the research, the fact that many providers call vital flow an amount of water provided for free as a principle, not as an alternative to absolute disconnection caught our attention. This is the case in Colombia. Italy applies both approaches: a minimum volume free for poor households (50 lpcd), and when technically feasible, flow restrictors in alternative to water shutoff, also providing 50 lpcd. The Italian regulatory authority (ARERA) implemented a new water default norm in 2021, which sets water supply disconnection procedures among other rules and statements [32,41].

Although Colombia Constitution has not recognized water as a human right explicitly, the Constitutional Court considered it throughout its judgments. People protected by Colombian law cannot be disconnected from the water services, and they must have access to a minimum water flow for health maintenance [30]. At the same time, the court recognized the utility's right to the revenue and payment mutual agreements. Some

Colombian cities have implemented a minimum free water flow policy for poorer families. The benefit is automatic, and the subsidized volume varies (monthly, 2.5 m$^3$/person in Medellin and 6 m$^3$/household in Bogotá). Again, this policy is not an alternative to water shutoffs, but an approach to guarantee affordable vital water access. Municipalities fund those programs, and it has also been implemented in other cities, such as Bucaramanga, Cali, Pereira, and Manizales.

**Table 3.** Case study disconnection policies.

| Case Studies | Service Disconnection Allowed? | Disconnection Restrictions | Vital Water Flow? |
|---|---|---|---|
| Australia | No | Service disconnection is not allowed in an occupied household. | Yes |
| Brazil | Yes | The water utility must communicate with the user about the upcoming disconnection. | Pilot tests in four towns |
| Colombia | Yes | The court forbids service disconnection to people protected by law. | Yes [1,2] |
| France | No | Service disconnection is possible for non-residential users, and for second properties. | No |
| Germany | Yes | The water utility must communicate with the user about the upcoming disconnection. | No |
| Italy | Yes | The water utility must communicate with the user about the debts. If technically feasible, the provider can reduce the water flow to 50 lpcd. | Yes, if technically feasible |
| Mexico | Yes | - | No |
| Portugal | Yes | The water utility must communicate to the user all possible actions to avoid the upcoming disconnection. | No |
| Singapore | Yes | The water utility notifies the user before disconnecting. | No |
| Spain | Yes | The water utility must visit the user before interrupting the services. The majority of disconnections occur with non-residential users. | No |
| UK | No | Service disconnection is allowed just for non-residential users, and second properties. Hospitals, schools, orphanages, and elderly homes cannot be disconnected. | No |
| Uruguay | Yes | The provider must communicate with the user about the upcoming disconnection. | No |

[1] Some cities apply a free first water block as a subsidy, called the "minimum vital water flow". [2] There is one pilot project of flow restriction implemented by one water provider.

Although water default rates are very low in Australia, disconnection for an occupied household is not permitted [42,43]. Regarding the minimum water flow policy, the utilities Sidney Water and Hunter Water implemented flow reduction after a debt notification as a way to avoid water shutoffs. The national Mexican normative includes service disconnection as one possible practice for users in debt, but a 50 lpcd for basic needs must be provided with other sources, such as water trunks or hydrants.

The Portuguese law determined disconnection rules, allowing utilities to use this practice, but with some restrictions, such as a previous warning to debtors and a 20-day gap between warning and disconnection. Although it is allowed to disconnect debtors in Spain, the initiative is not common due to a concern of correctly distinguishing between clients who are not willing to pay and users who are not able to pay. It is worth mentioning that water disconnection to poorer families is not allowed, especially if identified by social local authorities [23]. Uruguay was the first country in the world to recognize WSS as a human right, but allows water disconnection for debtors, according to the national

water provider regulation no. 157/10. The French law, Ley Brottes no. 312/2013, on the other hand, forbids WWS disconnections to residential users, no matter the user's economic situation [23]. In addition, the French Court has forbidden water flow restriction to residential users by utilities. Disconnections are allowed for commercial, industrial, government, and second-house users.

The Water Industry Act (1999) in England and Wales does not allow water disconnection for residential users, even in case of non-payment. To disconnect a client who is not going through financial problems, the utility must prove it to the court. Scotland does not allow water disconnection for residential users, and since residential users do not pay for WWS in North Ireland, they cannot be disconnected either. Brazil allows the disconnection, with a previous warning to debtors and at least 30 days for the payment, according to the national law, no. 11.445/2007. Currently, four cities in Paraná and São Paulo states are under pilot projects for water flow reduction through equipment installed on the water pipe, just right before the meter.

Although water flow restrictors are a humane alternative to complete disconnection, it was observed that this approach is very limited. Technical issues may be barriers to a broader application of minimum water flow policies as an alternative to water shutoffs. For example, in Italy, France, and Brazil, one single meter is commonly used for an entire building, making the disconnection of few debtors very difficult to implement. Individual metering is mandatory in Brazil for new buildings, according to the national law, no. 13.312/2016 [44]; however, older ones still face the problem mentioned before, which makes the application of water flow restrictors impossible.

The policies and practices observed from the case studies show that high-income countries have prioritized social assistance over water disconnection due to the serious health consequences the latest approach carries out. Even countries that allow WSS disconnection, such as Italy, Spain, Germany, Portugal and Singapore, impose several restrictions to do so, and offer ways for affordable WSS to vulnerable families. The fact that those countries have a higher GDP, less people in poverty, and 100% (or almost) WSS coverage rates, may enable governments to improve and to better target their assistance programs. On the opposite end, the case studies from Latin America show that despite some local social broader initiatives, such as ones applied by major Colombian cities, the disconnection policies are the main activity used to decrease default rates. We also observed the existence of a great fragmentation of WWS assistance programs and policies, raising local autonomy for default rate reduction policies. Even though local socioeconomic characteristics are essential to design proper policies, this lack of cohesion weakens successful approaches to guarantee water access to the poorest families. Some countries, such as Mexico and Brazil, have potential to improve their WSS access targeting the subsidies for those with actual needs, since they are medium-income nations, and suffer from great inequality. An integrated and coordinated financial public policy between the main entities—providers, regulators, and government—should definitely contribute to raise coverage and affordability indicators.

This paper has some limitations regarding the data availability. All the information gathered is from public documents, such as utilities and international organization reports, local laws and regulation, and they can be modified by the time of publication, since local elections happened in the last year in some of the countries analyzed.

## 4. Conclusions

Water and wastewater access is a concern worldwide, and it has become a crucial topic after the pandemic. This study gathered information about different subsidy policies applied in various socioeconomic contexts in eleven case studies around the globe, aiming to understand how utilities and governments are dealing with water default rates and service disconnections, and to inspire other utilities and governments that face the same issue.

We observed that WWS average price is 126% higher in high-income countries than in Latin American medium-income ones. At the same time, the water default rate is quite lower in richer nations, varying between 0.5% and 2% (except south Italy). Considering the

high-income countries in this study, half offer social tariffs as a way to support vulnerable families. Brazil, Mexico, and Colombia also provide this benefit to vulnerable households. In three nations, despite the absence of social tariffs, there is a broader social assistance for poorer families through general funds.

Regarding WWS disconnection policies, we concluded that their application is allowed in the majority of the countries studied here, regardless of income level. Despite this fact, the restrictions for implementing the WSS disconnection are greater in high-income countries, in order to protect vulnerable families' access. The United Kingdom, Australia and France even forbid water disconnection to residential users. The case studies with a bigger proportion of people facing poverty, affordability issues, or even lacking WSS, are the ones where disconnection policies have fewer restrictions. Since this paper did not observe a robust, coordinated, and broader approach to fighting infrastructure affordability concerns in medium-income countries, it may appear that utilities play a much more important role in those cases, leaving fewer alternatives to maintain service quality with revenue from tariffs. Minimum water flow initiatives are still very limited. Only Australia, Italy (when technically feasible), and few cities in Brazil and Colombia have some practices or tests in place. Additionally, as mentioned before, France, Italy and Brazil face the challenge of one single meter for the whole building, especially in poorer neighborhoods.

Finally, we observed that social assistance programs have been implemented over disconnection policies in high-income countries due to the high health risks the water restriction brings for humans. Those practices reveal that localities with a higher GDP and a smaller quantity of people struggling to pay WSS bills, are also the ones relying mainly on debt avoidance itself. Although water flow restriction is not spread, it could be a potential alternative to disconnection in order to guarantee the human right to WWSs, without jeopardizing utility economic and financial health.

**Author Contributions:** Conceptualization, R.C.M.; methodology, P.S. and I.M.; investigation, P.S. and I.M.; writing—original draft preparation, T.F.; writing—review and editing, R.C.M.; supervision, R.C.M. All authors have read and agreed to the published version of the manuscript.

**Funding:** This work is part of the research activity carried out at Civil Engineering Research and Innovation for Sustainability (CERIS) and has been funded by Portuguese Foundation for Science and Technology, under the project 2022.13852.BD.

**Institutional Review Board Statement:** Not applicable.

**Informed Consent Statement:** Not applicable.

**Data Availability Statement:** Not applicable.

**Conflicts of Interest:** The authors declare no conflict of interest.

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
