# Peer review of "Water Disconnection and Vital Flow Policies: International Practices in Medium- and High-Income Countries"

_water, doi:10.3390/w15050935_

Round 1

Reviewer 1 Report

The work is interesting in terms of water and sewage management problems. In the face of the Covid 19 pandemic and the energy crisis, they revealed themselves problems related to access to water supply services. Financial problems of water supply companies force savings. The problem of limiting supplies and excluding consumers is important from the point of view of human rights. The work presents an analysis of this problem in different countries. I believe that the work can be published with minor comments. In the abstract, please highlight the novelty and originality of the research. In the research methodology, please describe the analyzed systems in detail. Can the available data be subject to error? What should be the policy of the States in the aspect of water and sewage management costs? What is the relationship between the price for water supply services and the unit water consumption? Can this relationship be related to the quality of life indicator?  

Author Response

We appreciate all the comments from both reviewers to the Manuscript ID 2211491. They are all useful and helped us to clear the messages we would like to give. Here are the answers for each comment.

Response to reviewer #1

Comments:

The work is interesting in terms of water and sewage management problems. In the face of the Covid 19 pandemic and the energy crisis, they revealed themselves problems related to access to water supply services. Financial problems of water supply companies force savings. The problem of limiting supplies and excluding consumers is important from the point of view of human rights. The work presents an analysis of this problem in different countries. I believe that the work can be published with minor comments. In the abstract, please highlight the novelty and originality of the research. In the research methodology, please describe the analyzed systems in detail. Can the available data be subject to error? What should be the policy of the States in the aspect of water and sewage management costs? What is the relationship between the price for water supply services and the unit water consumption? Can this relationship be related to the quality of life indicator?  

Response:

The abstract was improved in order to include the main research contributions. There are also details between lines 143-158.

The data came from official reports, including local laws and regulations, and can be modified considering local elections. This limitation was added on the end of Section 3.

The financial public policy for WSS is complex and involves local and different perspectives, socioeconomic conditions and all the multiple goals the infrastructure sector carries out[1] Nevertheless, an opinion based on the case studies and on the paper topic was added on Section 3. 

The relationship between the price for water and consumption is known as price elasticity, and for residential consumers, the price elasticity of demand is found to be less than one, sometimes close to zero[2]. Nevertheless, in this paper, as the goal was not to analyze this subject, the sample is not enough to run the statistical analysis.  

[1] Pinto, F.S.; Marques, R.C. New Era / New Solutions: The Role of Alternative Tariff Structures in Water Supply Projects. Water Res 2017, 126, 216–231

[2] Nauges, C., Whittington, D., 2010. Estimation of water demand in developing countries: an overview. World Bank Res. Observer 25 (2), 263-294. Arbués, F., García-Valiñas, M.A., Martínez-Espiñeira, R., 2003. Estimation of residential water demand: a state of the art review. J. Socio-Economics 32 (1), 81-102.

Reviewer 2 Report

This paper addresses an interesting and important issue happen in both high- and medium-income countries, that is water disconnection, and how flow policies can be applied to ensure the fundamental right to access water. This paper reads well and the chart illustrations are easy to follow. I have a few comments regarding the method and presentation of results.

1. The authors claimed that "this study aims to gather information on water disconnection and vital flow policies applied among several medium and high-income countries". To me, this is less of an objective, but more of a task. There is a lack of clear objective/goal for this study. What is the purpose of gathering this information?

2.  In section 2, the authors wrote, "The study was based on official documents, and interviews with the mains stakeholders, such as utilities, water associations, and regulatory agencies." I'd like to see more details about the interviews, what questions are asked? How many people were interviewed in each country? How many from each entity? Is the sample representative? How are answers analyzed? What approach was adopted to derive findings? I think this has to be done properly in writing. Only saying interviews are done with experts are not enough. 

3. In results, the authors wrote, "the most common way is charging wealthier groups more than the actual cost in order to reduce the price for the vulnerable population". I'd like to see reference for this claim as well as some discussion around the controversies surrounding it. I think some of the statements were done in a hand-waving fashion without giving much in-depth discussion about the implications. I'd like to see more elaborations on the implications of current flow policies and potential issues of suggested alternative solutions. The paper in its current form reads like a compilation of information instead of a well-articulated discussion paper with the authors' own insights. I think having more in-depth discussion will greatly strengthen the impact and value of this paper.

Author Response

We appreciate all the comments from both reviewers to the Manuscript ID 2211491. They are all useful and helped us to clear the messages we would like to give. Here are the answers for each comment.

Comments:

This paper addresses an interesting and important issue happen in both high- and medium-income countries, that is water disconnection, and how flow policies can be applied to ensure the fundamental right to access water. This paper reads well and the chart illustrations are easy to follow. I have a few comments regarding the method and presentation of results.

1.The authors claimed that "this study aims to gather information on water disconnection and vital flow policies applied among several medium and high-income countries". To me, this is less of an objective, but more of a task. There is a lack of clear objective/goal for this study. What is the purpose of gathering this information?

2.In section 2, the authors wrote, "The study was based on official documents, and interviews with the mains stakeholders, such as utilities, water associations, and regulatory agencies." I'd like to see more details about the interviews, what questions are asked? How many people were interviewed in each country? How many from each entity? Is the sample representative? How are answers analyzed? What approach was adopted to derive findings? I think this has to be done properly in writing. Only saying interviews are done with experts are not enough. 

  1. In results, the authors wrote, "the most common way is charging wealthier groups more than the actual cost in order to reduce the price for the vulnerable population". I'd like to see reference for this claim as well as some discussion around the controversies surrounding it. I think some of the statements were done in a hand-waving fashion without giving much in-depth discussion about the implications. I'd like to see more elaborations on the implications of current flow policies and potential issues of suggested alternative solutions. The paper in its current form reads like a compilation of information instead of a well-articulated discussion paper with the authors' own insights. I think having more in-depth discussion will greatly strengthen the impact and value of this paper.

Response:

Item 1) The abstract was improved (lines 17-21).

Item 2) An explanation was added to Section 2. One representative of each entity was designed to answer us. Taking into account that the interviews had the purpose of checking available data or missing essential documents regarding subsidies, disconnection, and vital flow policies, we did not run a statistical analysis on the sample representativeness. The answers - additional documents or practices in place - were analyzed and discussed qualitatively within the paper.

Item 3) Reference about cross-subsidization included between lines 203-213.

Discussion and insights were added in Sections 3 and 4. It is not the intention of this paper to propose own suggestions of best solutions (since they can vary according to local socioeconomic, political, and cultural characteristics), but to provide a discussion about the variety of policies applied according to different socioeconomic conditions. 

An English revision was carried out.

Once again, we thank the reviewers’ comments and recommendations.